# Impact of COVID-19 on household food insecurity and interlinkages with child feeding practices and coping strategies in Uttar Pradesh, India: a longitudinal community-based study

Phuong Hong Nguyen  ,[1] Shivani Kachwaha,[2] Anjali Pant,[2] Lan M Tran,[3] Sebanti Ghosh,[4] Praveen Kumar Sharma,[4] Vishal Dev Shastri  ,[4] Jessica Escobar-Alegria,[5] Rasmi Avula,[2] Purnima Menon[2]

For numbered affiliations see end of article.

**Correspondence to**
Dr Phuong Hong Nguyen;
P.H.Nguyen@cgiar.org

## ABSTRACT

**Objectives** The COVID-19 pandemic has profound negative impacts on people's lives, but little is known on its effect on household food insecurity (HFI) in poor setting resources. This study assessed changes in HFI during the pandemic and examined the interlinkages between HFI with child feeding practices and coping strategies.

**Design** A longitudinal survey in December 2019 (in-person) and August 2020 (by phone).

**Setting** Community-based individuals from 26 blocks in 2 districts in Uttar Pradesh, India.

**Participants** Mothers with children <2 years (n=569).

**Main outcomes and analyses** We measured HFI by using the HFI Access Scale and examined the changes in HFI during the pandemic using the Wilcoxon matched-pairs signed-rank tests. We then assessed child feeding practices and coping strategies by HFI status using multivariable regression models.

**Results** HFI increased sharply from 21% in December 2019 to 80% in August 2020, with 62% households changing the status from food secure to insecure over this period. Children in newly or consistently food-insecure households were less likely to consume a diverse diet (adjusted OR, AOR 0.57, 95% CI 0.34 to 0.95 and AOR 0.51, 95% CI 0.23 to 1.12, respectively) compared with those in food-secure households. Households with consistent food insecurity were more likely to engage in coping strategies such as reducing other essential non-food expenditures (AOR 2.2, 95% CI 1.09 to 4.24), borrowing money to buy food (AOR 4.3, 95% CI 2.31 to 7.95) or selling jewellery (AOR 5.0, 95% CI 1.74 to 14.27) to obtain foods. Similar findings were observed for newly food-insecure households.

**Conclusions** The COVID-19 pandemic and its lockdown measures posed a significant risk to HFI which in turn had implications for child feeding practices and coping strategies. Our findings highlight the need for further investment in targeted social protection strategies and safety nets as part of multisectoral solutions to improve HFI during and after COVID-19.

## Strengths and limitations of this study

► The longitudinal study design allowed measuring the impact of COVID-19 on household food insecurity (HFI) and its implications for child feeding practices and coping strategies in the context of low-income and middle-income countries.

► The study demonstrated the feasibility of measuring HFI via digital data collection methods but indicated some challenges including low response rate and potential response bias.

► The sample of mothers with children <6 months constrained comparison of child feeding practices before and during COVID-19.

► A single point 24-hour dietary recall may be unrepresentative of child feeding patterns.

► The study was not able to assess whether the increase in level of HFI affected child growth.

## INTRODUCTION

The COVID-19 has profound and wide-ranging public health impacts and poses a significant global threat to development. Beyond the direct impacts from the virus, the pandemic will likely have a range of indirect consequences on food insecurity, child malnutrition, morbidity and mortality through disruptions in health and nutrition services, food supply chains and livelihoods.[1–4] Early estimates suggest that potential disruptions of health systems and decreased access to food could lead to 1 157 000 additional child deaths and 56 700 additional maternal deaths.[5] Further, disruptions caused by the pandemic may affect households in multiple other ways including employment and income loss, mobility constraints and household stress. Experts have warned about the potential consequences of COVID-19,

ruining decades of progress, making it unlikely for low-income and middle-income countries (LMICs) to reach the sustainable development goal to 'end hunger, achieve food security and improved nutrition and promote sustainable agriculture' by 2030.[6]

There have been growing concerns on the impact of COVID-19 on household food insecurity (HFI).[7 8] Disruptions caused by the pandemic have the potential to influence all 'four pillars' of food security including availability, access, utilisation and stability.[9] The pandemic may influence HFI directly on the supply side by disrupting food systems (such as primary food production, stability of food production, processing, food reserve stockpiles and marketing) as well as indirectly on the demand side due to impact of lockdowns on households' incomes, physical access to food and economic access to food.[10 11]

The impact of COVID-19 on HFI and poor health outcomes is complex, multilevel and bidirectional.[4] At the household and individual levels, food insecurity is hypothesised to be a risk factor for both short-term and long-term health outcomes through key three pathways: household stress (due to worrying about health issues, job loss and strained finances and disconnection from social support systems), behavioural coping mechanisms (engaging in high-risk behaviour, compromising healthcare activities for foods, poor mental health and inadequate child feeding and nurturing) and inflammatory pathways.[4] Expected negative consequences on food, nutrition and health security of vulnerable groups including young children, pregnant and lactating women may further exacerbate existing social and health inequities.[12]

Despite established frameworks and global understanding of the threat to HFI during the pandemic, empirical investigations are very limited to date. Available information on HFI was mainly collected during the pandemic[13–16] and very few studies have examined the dynamic changes of HFI over the COVID-19 pandemic's evolution in LMICs,[17 18] particularly in the South Asian or Indian context. India is facing a double crisis—COVID-19 and food insecurity,[19] carrying the second highest burden of COVID-19 in the world with nearly 8 million total confirmed cases and 119 502 deaths as of 28 October 2020.[20] Yet only few studies are available on food security using data at the farmer and supply-side level,[21 22] and negligible evidence on the demand side. Very little is known about how women and children within households may be affected by HFI. Further, there is lack of empirical evidence on the changes in HFI during the pandemic. Addressing this knowledge gap is critical for action, specifically at this decisive time in India when the COVID-19 trajectory is still uncertain, and there is concern about potential spikes in the coming months. Our study seeks to address this gap in the current literature with the objectives to (1) assess the changes in HFI before and during the pandemic in Uttar Pradesh, India and (2) examine the interlinkages between HFI with child feeding practices and coping strategies to deal with household economic hardships and obtain foods.

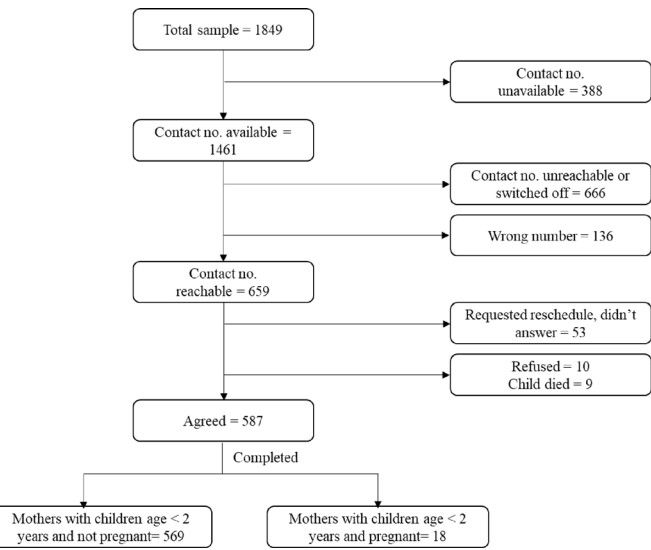

**Figure 1** Participant flow.

## METHODS

### Design

This study is a follow-up of a cluster randomised trial (2017–2019), which aimed to assess the impact of strengthening delivery of maternal nutrition interventions, including micronutrient supplements and intensifying interpersonal counselling and community mobilisation, implemented through government antenatal care (ANC) services in Uttar Pradesh, India.[23] Details of the parent study have been described elsewhere.[24] Briefly, we conducted in-person repeated cross-sectional surveys of 1800 recently delivered women as part of the cluster randomised trial.[23] The endline data collection was conducted in December 2019, prior to the onset of COVID-19 pandemic, providing an opportunity for a pre-assessment and post-assessment of the effect of COVID-19 on food insecurity in this context.

### Data sources

The household survey was conducted with mothers of children <2 years old following the same study design and sampling frame as in the cluster randomised trial. Of the 1849 mothers surveyed at endline from the parent study in December 2019, 587 could be reached for a phone interview in August 2020, yielding a response rate of 32% (figure 1). Reasons for not being able to conduct phone survey included unavailable phone number (n=388), phone unreachable or switched off (n=667), wrong number (n=136), refusal to participate (n=63) and child death (n=9). Reasons for lost to follow-up in the phone survey were similar between intervention and comparison areas (results not shown). The total sample of non-pregnant mothers (n=569) interviewed in both surveys were used for the analysis.

### Variables

Household food security was measured before (in-person) and during the pandemic (by phone) using the standard Food and Nutrition Technical Assistance Project/United

States Agency for International Development (FANTA/USAID) HFI Access Scale.[25] A recent study in Mexico examined the internal validity of food insecurity scales administered through in-person vs phone surveys and found phone surveys were a feasible strategy to measure food security during COVID-19.[18] Mothers were asked nine questions related to the household's experience of food insecurity in the 30 days preceding the survey. These questions capture three main domains of HFI: anxiety and uncertainty about the household food supply (one item), insufficient quality (three items) and insufficient quantity and its physical consequences (five items). We reported the percentage of households that experienced (1) any food insecurity occurrence among nine questions, (2) any of a specific domain and (3) food insecurity condition categorised as food secure and mild, moderately or severely food insecure.

Information on child feeding practices was assessed using the standard WHO indicators,[26] on the basis of the maternal recall of all foods and liquids consumed by the child in different time periods of the previous 24 hours before the survey (online supplemental table 1). All food items were categorised into the seven food groups used in the WHO guideline[26]: (1) starchy staple foods, (2) legumes and nuts, (3) dairy products (milk, yoghurt and cheese), (4) flesh foods, (5) eggs, (6) vitamin A rich fruits and vegetables, and (7) other fruits and vegetables. Minimum dietary diversity was defined as children who consumed foods from four or more out of seven food groups in the previous 24 hours. Data for complementary feeding practices were not available during the in-person survey in December 2019, because all mothers had children <6 months during that time.

Households were also asked about access to social protection, especially food supplementation they received for mothers and children from the government during the lockdown period and during the 30 days prior to the survey, such as take-home rations (THR) and use of public distribution system (PDS). Finally, information on different coping strategies that the household had to engage in the past 30 days due to lack of food was collected, including spending savings, reducing essential non-food expenditure, borrowing money or selling jewellery/gold.

Other potential factors associated with food security or child feeding practices were obtained for mothers (age, education level and occupation), child (age and sex) and households (religion, scheduled caste/tribal—designated historically disadvantaged groups in India, number of children <5 years and household socioeconomic status (SES)). The SES index (collected in person survey) was constructed using a principal component analysis from multiple variables including household ownership of assets, livestock and housing quality.[27]

## Data analysis

We compared background characteristics of the analytic sample (mothers who completed both surveys, in-person survey before COVID-19 and phone survey during COVID-19) and the non-analytical sample (those who completed in-person surveys only) using Student's t-test (for continuous variables) and $\chi^2$ test (for categorical variables). We used descriptive analysis to report HFI before and during the pandemic and child feeding practices. We examined changes in HFI before and during the pandemic using Wilcoxon matched-pairs signed-rank tests.

To examine differences in child feeding practices and coping strategies by food insecurity status, we created three categories of households: (1) food secure (households that were food secure before and during COVID-19 pandemic), (2) consistently food insecure (households that were food insecure before and during COVID-19) and (3) newly food insecure (households that were food secure before COVID-19 but became food insecure during the pandemic). We then compared child feeding practices and coping strategies among the three categories using multivariable regression models, adjusting for child age and sex, breastfeeding status, mother's age, religion, education, scheduled caste, number of children <5 years in the household and household SES. We also examined uptake of social protection programmes such as food supplementation and cash transfer as potential strategies to improve HFI. All statistical analyses were undertaken using Stata V.16. Statistical significance was defined as p<0.05.

## Patient and public involvement statement

Patients or the public were not involved in the design, or conduct, or reporting, or dissemination plans of our research.

## RESULTS

### Characteristics of the study sample

At the time of in-person survey in December 2019, all mothers had an infant between the ages of 0–5.9 months of age with an average age of 3 months (table 1). On average, mothers were ~26 years and the majority of them (>90%) were housewives. Nearly all women were Hindu (92%) and nearly half of them belonged to a backward community (44%–47%). Mothers in the final analytical sample had higher education (8.2 vs 6.7 years of schooling, p<0.001) and lived in wealthier (27 vs 17% in quintile 5, p<0.001) and more food secure households (79 vs 75%, p=0.08) compared with those in the non-analytical sample. Mothers belonging to intervention and control areas of the maternal nutrition intervention (from 2017 to 2019) were equally represented in the analytic sample. In the follow-up phone survey in August 2020—children were on average 11.6 months old (ranging between 8 and 14 months).

### Changes in food security status during the COVID-19 pandemic

Prior to the pandemic, 21% of households were identified as food insecure. Six months into the pandemic, the prevalence of any food insecurity increased from 21% to 80%, of which mildly, moderately and severely food

**Table 1** Background characteristics* of the study sample that participated in surveys before and during the COVID pandemic (December 2019 and August 2020)

| | Analytic sample (both in person and phone surveys before and during the pandemic) | Non-analytical sample (only in person survey before the pandemic) | P value |
|---|---|---|---|
| | (n=569) | (n=1280) | |
| Age of respondent mother (years) | 25.5 (3.8) | 25.7 (4.0) | 0.47 |
| Education (years) | 8.2 (4.3) | 6.7 (4.6) | <0.001 |
| Never attended school | 14.1 | 24.8 | <0.001 |
| Primary school (grade1-5) | 13.9 | 16.3 | |
| Middle school (grade 6–9) | 24.3 | 24.7 | |
| High school (grade 10–12) | 30.1 | 23.3 | |
| Graduate and above | 17.8 | 10.9 | |
| Occupation as housewife | 91.7 | 93.0 | 0.35 |
| Child age, months | 3.0 (1.6) | 2.8 (1.6) | 0.041 |
| Child sex (male) | 49.0 | 49.5 | 0.84 |
| No of children <5 years | 1.6 (0.7) | 1.7 (0.7) | 0.60 |
| Religion as Hindu | 93.7 | 91.1 | 0.061 |
| Caste category | | | |
| Scheduled caste/tribe | 38.3 | 38.4 | 0.25 |
| Other backward class | 44.3 | 47.0 | |
| General/others | 17.4 | 14.5 | |
| Household socioeconomic status | | | |
| Quintile 1 | 11.6 | 23.8 | <0.001 |
| Quintile 2 | 19.2 | 20.4 | |
| Quintile 3 | 18.1 | 20.9 | |
| Quintile 4 | 24.6 | 18.0 | |
| Quintile 5 | 26.5 | 17.0 | |
| Household food security status | | | |
| Food secure | 79.3 | 74.5 | 0.08 |
| Mildly food insecure | 5.6 | 5.9 | |
| Moderate food insecure | 5.1 | 5.3 | |
| Severe food insecure | 10.0 | 14.3 | |
| Maternal nutrition (2017–2019) | | | |
| Intervention area | 282 | 640 | |
| Comparison area | 287 | 640 | |

*Background data presented in this table were from in-person survey in December 2019.

insecure households increased by 14 percentage points (pp), 25 pp and 20 pp, respectively (figure 2A). Overall, 62% households changed from being food secure to food insecure during the pandemic. HFI experiences sharply increased for each domain. For example, the prevalence of anxiety and uncertainty about the household food supply, insufficient quality of food and insufficient

quantity of food consumed during the pandemic were 45%, 78% and 42%, respectively, which was much higher than before the pandemic (12%, 18% and 14%, respectively) (figure 2B).

## Child feeding practices during the COVID-19 pandemic

Child feeding practices are of major concern, with only 19% of children achieving minimum dietary diversity (≥4 food groups). An extremely low proportion of children were fed flesh foods (1%), eggs (1%) and vitamin-A rich fruits and vegetables (4%). One-third of the children consumed other fruits and vegetables and nearly two-thirds consumed legumes and nuts in the 24 hours prior to the survey (figure 3).

## Association between food insecurity status and child feeding practices during the COVID-19 pandemic

Children living in households that became food insecure since the pandemic were less likely to consume a diversified diet (18% vs 28%; adjusted OR, AOR 0.57, 95% CI 0.34 to 0.95) as well as legumes and nuts (57% vs 69%; AOR 0.61, 95% CI 0.38 to 0.97) compared with children living in consistently food-secure households (table 2). Child feeding practices were worse in the households that were food insecure at both times. Specifically, fewer children in consistently food-insecure households consumed a diverse diet (12.4% vs 28%; AOR 0.51, 95% CI 0.23 to 1.12) and other fruits and vegetables (21% vs 40%; AOR 0.50, 95% CI 0.26 to 0.97) compared with those in food-secure households.

## Challenges faced during the pandemic

The key challenges faced by households in consuming food in the last 7 days preceding the survey included non-availability of funds to buy food (59%), non-availability of foods in market area (21%), increase in food prices (17%) and inability to travel or transport issues (21%). The pandemic-related challenges had resulted in unemployment/loss of income in 78.4% households (figure 4).

## Coping strategies and HFI status during the COVID-19 pandemic

More than 60% of households disbursed their savings and reduced their expenses on health and non-food essentials to meet food and other requirements, irrespective of their food security status (table 3). Households experiencing food insecurity were more likely to engage in coping strategies related to obtaining food including reducing their expenditure on non-food essentials (AOR 1.7, 95% CI 1.08 to 2.78 and AOR 2.2, 95% CI 1.09 to 4.24 for newly and consistently food-insecure households, respectively), borrowing money to buy food (AOR 3.6, 95% CI 2.19 to 5.80 and AOR 4.3, 95% CI 2.31 to 7.95, respectively) and selling jewellery (AOR 3.0, 95% CI 1.16 to 7.92 and AOR 5.0, 95% CI 1.74 to 14.27, respectively). Additionally, newly food-insecure households were ~2 times more likely to spend saving or sell households/assets/transport means.

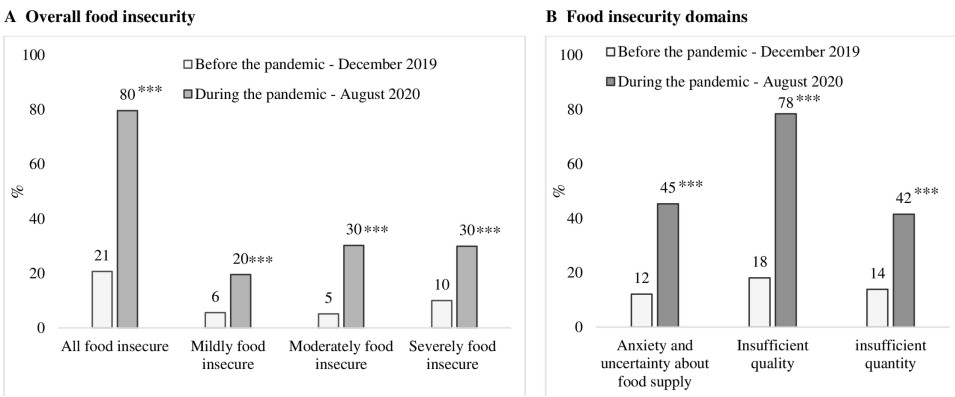

**Figure 2** Food insecurity experienced by mothers and their household members in the previous 30 days before and during the COVID-19 pandemic. Significant change from before and during the pandamic: ***p<0.001.

## Social protection before and during the COVID-19 pandemic

The proportion of households where children received THR from the Integrated Child Development Services (ICDS) programme was similar before and during the pandemic and was slightly higher in food-insecure (~63%) compared with food-secure households (55%–59%) (figure 5). Coverage of PDS rations increased significantly during the pandemic for both food-insecure (61% to 71%) and food-secure households (from 49% to 72%); the increase was smaller among beneficiaries from consistently food-insecure compared with those in food-secure households (9.3 pp vs 23 pp).

## DISCUSSION

In response to global concerns on the impact of COVID-19 on maternal and child food and nutrition insecurity, our study provides unique evidence of changes in HFI before and during the pandemic and its linkages with child feeding practices as well as coping strategies to obtain foods among food secure and insecure households. We found that HFI increased substantially during the pandemic (60 pp), with a large portion related to insufficient quality (78%) and lower levels related to insufficient quantity (42%). Children living in food-insecure households were less likely to consume a diversified diet, mainly due to less consumption of legumes and nuts, fruits and vegetables. In order to overcome the challenges during the pandemic, households were compelled to engage in several coping strategies related to spending existing savings, reducing household expenditures, selling assets or borrowing money.

Our findings were consistent with the global literature on the increase in HFI during the pandemic.[28–32] However, most previous studies mainly obtained information during the pandemic and did not have data prior to the onset of the pandemic. A rapid assessment conducted in LMICs including Kenya, Nigeria, Mozambique and Rwanda showed that 79%–87% of respondents were worried about lack of sufficient food during COVID-19.[13] Similarly, nearly 90% of households in rural and urban Bangladesh experienced different levels of food insecurity and engaged in financial or food compromised coping strategies.[15] The prevalence of moderate to severe HFI during the COVID-19 lockdown was lower in Peru, affecting 23% of households, with predictors being low income pre-pandemic, income reduction or running out of savings during the pandemic.[14] Among the few studies with information before and during COVID-19 time, two were from the USA, one found 32% increase in HFI since COVID-19[16] while the other found an increase of 20%.[28] Only two other studies provided estimates of HFI before and during COVID-19 where one found an increase of 14 pp (from 61.1% to 75.1%) in any HFI in Mexico[18] and the other observed an increase of 43.4 pp (from 8.3% to 51.7%) in moderate and severe HFI in Bangladesh.[17] Our study showed much higher magnitude of increase in HFI (~60 pp) compared with other studies, which is a worrisome finding given the high pre-existing levels of food insecurity in India. We also found that HFI was predominantly due to insufficient food quality concerns which was aligned with a previous study which showed increased consumption of high-calorie snack foods and sweets,[28 30] or cheaper highly processed foods.[4]

Our findings indicate challenges to several food security dimensions, including livelihood and income loss, economic and physical access, availability and utilisation. A study on livelihood and dietary effects of COVID-19 with vegetable producers in four states of India reported negative impacts

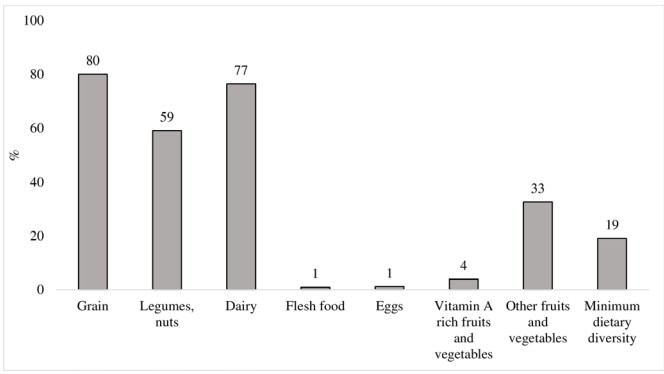

**Figure 3** Child feeding practices during the COVID-19 pandemic.

**Table 2** Association between child dietary diversity and household food insecurity status during the pandemic

| | Currently food secure | Newly food insecure* | Consistent food insecurity | New food insecurity versus food secure† | | | | Consistent food insecurity versus food secure† | | | |
|---|---|---|---|---|---|---|---|---|---|---|---|
| | n=116 | n=354 | n=99 | | | Adjusted OR | | | | Adjusted OR | |
| | % | % | % | Crude OR (95% CI) | P value | (95% CI) | P value | Crude OR (95% CI) | P value | (95% CI) | P value |
| Grain | 79.3 | 80.8 | 78.8 | 1.1 (0.65 to 1.85) | 0.73 | 0.98 (0.57 to 1.69) | 0.95 | 0.97 (0.50 to 1.87) | 0.93 | 0.87 (0.43 to 1.77) | 0.64 |
| Legumes and nuts | 69.0 | 56.8 | 55.6 | 0.59 (0.38 to 0.93) | 0.02 | 0.61 (0.38 to 0.97) | 0.04 | 0.56 (0.32 to 0.98) | 0.04 | 0.69 (0.38 to 1.25) | 0.22 |
| Dairy | 74.1 | 76.3 | 79.8 | 1.12 (0.69 to 1.82) | 0.64 | 1.22 (0.74 to 2.01) | 0.43 | 1.38 (0.72 to 2.62) | 0.33 | 1.72 (0.87 to 3.41) | 0.12 |
| Flesh foods | 0.9 | 0.6 | 2.0 | 0.66 (0.06 to 7.36) | 0.74 | 0.63 (0.05 to 7.47) | 0.72 | 2.37 (0.21 to 26.55) | 0.48 | 1.46 (0.09 to 23.2) | 0.79 |
| Eggs | 0.9 | 1.1 | 1.0 | 1.33 (0.15 to 12.02) | 0.80 | 1.10 (0.11 to 10.5) | 0.94 | 1.17 (0.07 to 19.01) | 0.91 | 0.87 (0.04 to 17.0) | 0.93 |
| Vit A rich fruits and vegetables | 4.3 | 4.3 | 2.0 | 0.99 (0.35 to 2.79) | 0.99 | 0.77 (0.26 to 2.26) | 0.64 | 0.46 (0.09 to 2.41) | 0.36 | 0.31 (0.05 to 1.79) | 0.19 |
| Other fruits and vegetables | 39.7 | 33.5 | 21.2 | 0.77 (0.50 to 1.18) | 0.17 | 0.73 (0.46 to 1.16) | 0.18 | 0.41 (0.22 to 0.75) | 0.004 | 0.50 (0.26 to 0.97) | 0.042 |
| Minimum dietary diversity (≥4 food groups) | 28.1 | 17.9 | 12.4 | 0.56 (0.34 to 0.91) | 0.02 | 0.57 (0.34 to 0.95) | 0.03 | 0.36 (0.17 to 0.75) | 0.006 | 0.51 (0.23 to 1.12) | 0.09 |

*Currently food secure was defined as households who were food secure before and during COVID-19 pandemic and those who were food insecure at some point before but were no longer food insecure during the pandemic; consistent food insecurity was defined as both food insecure before and during COVID-19; newly food insecurity insecure was defined as food secure before COVID-19 but became food insecure during the pandemic.

†Model controlled for child age, sex, breastfeeding status, mother's age, education, caste, religion, number of children <5 years and household SES.

SES, socioeconomic status.

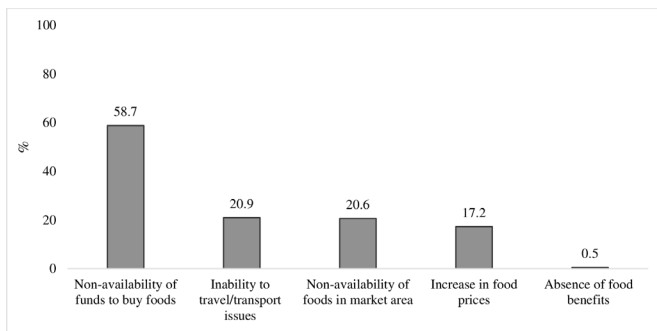

**Figure 4** The key challenges faced by households during the COVID-19 pandemic.

on production, sales, prices, and incomes among majority of farmers,[21] Farm households also reported disruptions to their diets with reduced ability to access nutrient-dense foods, particularly fruit and animal source foods.[21] Another study in Maharashtra, India found disruptions in the urban–rural food supply chain due to the closure of wholesale markets with uncertainties in food supply, declines in market availability and increase in food prices.[22] These findings are complementary to our study and the supply-side insights possibly explain some of the trends we observe in food security, child feeding and coping strategies.

To our knowledge, infant and child feeding practices during the pandemic have not been explored in the literature. Our findings showed that the diets of children were suboptimal, with only 19% achieving minimum dietary diversity—a similar result compared with a previous study in Uttar Pradesh, India before COVID-19 pandemic (17%).[33] We also found that children living in food-insecure households had much poorer diets than those in food-secure households, but the proportion of children consuming flesh foods, eggs and vitamin A fruits and vegetables is very low, irrespective of food security status. During the COVID-19 pandemic, child feeding practices have been reported to change, particularly among food-insecure households, due to higher levels of stress, fewer resources and less access to food and affordability, leading to restrict the quantity and quality of food their children eat and more parents' controlling feeding behaviours.[28] Other studies also showed that mothers of the children in food-insecure households often prioritised shelf-stable foods to deal with food supply disruptions and social-distancing policies, and have a tendency to rely on energy-dense foods for a longer period of time.[4]

We found that all households in our study engaged in some coping strategies to obtain food regardless of HFI status, but food-insecure households were more likely to engage in several such practices. Our findings are consistent with literature stating that the main strategies food-insecure households generally rely on to maintain access to food include shifting within their own spending patterns to prioritise food (reducing expenses on health, other non-food expenditures or agricultural, livestock or fisheries inputs), relying on social network or access to government nutrition programmes.[4 16] However, all these strategies can easily be impacted when COVID-19 pandemic severely affects the entire household

budget, or social-distancing policies could affect network access. The coping strategies households have adopted to obtain food during COVID-19 will run out and will not suffice for preventing HFI from getting worse if the pandemic crisis continues.

Social protection strategies are an important intervention to address the rising levels of HFI in the context of COVID-19, particularly for low-income countries.[34] A global review of evidence by the World Bank found that India increased coverage of cash transfers from ~2% before the pandemic to about 15% during COVID-19.[35] The Indian government also initiated home delivery of THR for pregnant and lactating women and children and provided 1-month free supply of wheat and rice to the poorest ration card holders through the PDS.[36] Our findings on the increased access to PDS during COVID-19 align with previous conclusions about the important role of the programme as an essential component of the Government's response to food insecurity.[37] Despite these measures, food supplementation was received among just over half of households and the increase in access to PDS was smaller among beneficiaries that are consistently food insecure compared with the food secure. These results highlight an important opportunity to strengthen the government's response to reduce food insecurity during and after COVID-19 in the short term by improving efficiency of existing social protection strategies and targeting to the most vulnerable populations.[37 38] A recent costing study conducted in Mexico found it would cost less than 0.06% of the gross domestic product to effectively safeguard families with young children through a cash transfer and basic services subsidy.[39] Other strategies which may be considered include outlining specific recommendations to ensure food security for poor and vulnerable populations as done for other LMICs in Africa[40] and include special initiatives for migrant populations.[41] Certain agricultural reforms may also be considered[42] such as home gardening,[43] diversification of production and strong local market chains[44] to alleviate HFI, improve diets and reduce reliance on coping strategies due to food insecurity.

Our study followed the cohort of mothers before the pandemic and 6 months after the onset of COVID-19, thus offering a unique and timely contribution to the literature on the magnitude and nature of increase in HFI before and during the pandemic, and its implications for child feeding practices and coping strategies in the context of LMICs with prevailing high HFI. Given the restrictions on movement and contacting people, we were able to mobilise the phone survey to reach mothers and use the same instrument to measure food security over time. Our experience demonstrated the feasibility of gathering information on HFI via digital data collection methods but indicated potential challenges and bias in the background characteristics of respondents interviewed through in-person versus phone surveys. Mothers who responded to phone survey had slightly higher education and SES background compared with non-responders, indicating that we may not be able to reach some of the poorest or most vulnerable households through phone surveys. We also experienced similar challenges as other phone surveys[45]

**Table 3** Association between current coping strategies and household food insecurity status during the pandemic

| | Currently food secure* | Newly food insecure* | Consistent food insecurity | New food insecurity vs food secure† | | | | Consistent food insecurity vs food secure† | | | |
|---|---|---|---|---|---|---|---|---|---|---|---|
| | n=116 | n=354 | n=99 | | | | | | | | |
| | % | % | % | Crude OR (95%CI) | P value | Adjusted OR (95%CI) | P value | Crude OR (95%CI) | P value | Adjusted OR (95%CI) | P value |
| Spent savings | 83.6 | 91.0 | 89.9 | 1.97 (1.07 to 3.63) | 0.03 | 2.05 (1.09 to 3.88) | 0.027 | 1.74 (0.77 to 3.95) | 0.18 | 1.73 (0.71 to 4.18) | 0.23 |
| Reduced health expenditure | 64.7 | 72.0 | 74.7 | 1.41 (0.90 to 2.20) | 0.13 | 1.33 (0.84 to 2.10) | 0.23 | 1.62 (0.90 to 2.92) | 0.11 | 1.49 (0.79 to 2.80) | 0.22 |
| Reduced other essential non-food expenditures such as education and clothes | 66.4 | 77.4 | 81.8 | 1.73 (1.10 to 2.74) | 0.02 | 1.73 (1.08 to 2.78) | 0.024 | 2.28 (1.20 to 4.32) | 0.01 | 2.15 (1.09 to 4.24) | 0.027 |
| Borrowed money to buy food | 25.0 | 54.8 | 63.6 | 3.64 (2.27 to 5.82) | <0.001 | 3.57 (2.19 to 5.80) | <0.001 | 5.25 (2.92 to 9.44) | <0.001 | 4.29 (2.31 to 7.95) | <0.001 |
| Reduced expenses on agricultural, livestock or fisheries inputs | 23.3 | 33.3 | 35.4 | 1.65 (1.02 to 2.67) | 0.04 | 1.64 (0.99 to 2.72) | 0.055 | 1.80 (0.99 to 3.27) | 0.05 | 1.78 (0.94 to 3.38) | 0.078 |
| Selling jewellery/gold | 4.3 | 13.0 | 21.2 | 3.32 (1.28 to 8.56) | 0.01 | 3.03 (1.16 to 7.92) | 0.024 | 5.98 (2.16 to 16.53) | 0.001 | 4.98 (1.74 to 14.27) | 0.003 |
| Selling household goods or productive assets or means of transport | 19.0 | 29.4 | 27.3 | 1.78 (1.06 to 2.98) | 0.03 | 1.78 (1.03 to 3.07) | 0.038 | 1.6 (0.84 to 3.04) | 0.15 | 1.64 (0.83 to 3.26) | 0.16 |

*Currently, food secure was defined as households who were food secure before and during COVID-19 pandemic and those who were food insecure at some point before but were no longer food insecure during the pandemic; consistent food insecurity was defined as both food insecure before and during COVID-19; newly food insecure was defined as food secure before COVID-19 but became food insecure during the pandemic.
†Model controlled for mother's age, education, caste, religion, number of children <5 years and household SES.
SES, socioeconomic status.

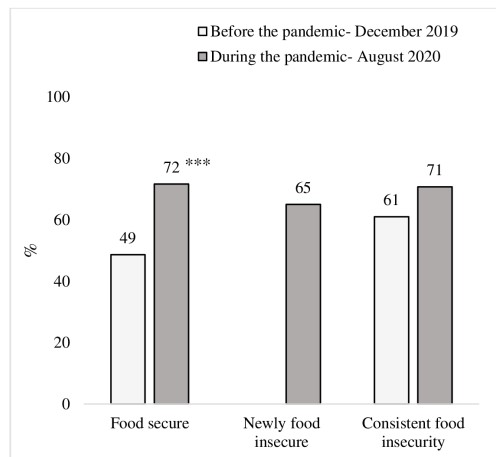

**Figure 5** Household receipt of social protection benefits before and during the pandemic, by household food insecurity status. Significant change from before and during the pandamic: ***p<0.001.

including low response rate, several calling schedules during the survey and potential unknown response bias or residual confounding factors. Since all mothers in our study had children <6 months in December 2019, we were unable to obtain information on complementary feeding to compare child feeding practices before and during COVID-19 time. Child feeding was assessed by a single point 24-hour dietary recall which may be unrepresentative of overall dietary exposure. Finally, we were not able to assess whether the increase in level of HFI affected child growth which should be considered in future research.

## Conclusion

The COVID-19 pandemic and lockdown measures arising from the pandemic had a significant negative impact on HFI in this context, which in turn had implications for child feeding practices and reliance on coping strategies to obtain foods. Our study highlighted the opportunity to reduce HFI in the short term with existing resources by improving the targeting of social protection benefits to effectively reach the food insecure and make quality diets accessible. Given the great concerns about the expected increase in HFI as the pandemic continues, strengthened multisectoral responses are needed to ensure effective re-establishment of health and nutrition services, food-supply chains and restoration of livelihoods to improve household food security during and after the pandemic. Policies response to the pandemic also require coordination across different governance systems to guide threat against HFI in future pandemics because the most important impact on food security is related to a serious slowdown in economic activity and disrupted supply chains caused by strict lockdown measures, not the pandemic itself.

## Author affiliations

[1]Poverty, Health and Nutrition Division, International Food Policy Research Institute, Washington, District of Columbia, USA
[2]Poverty, Health and Nutrition Division, International Food Policy Research Institute, New Delhi, India
[3]Alive & Thrive, FHI Solutions, Hanoi, Vietnam
[4]Alive & Thrive, FHI Solutions, New Delhi, India
[5]Alive & Thrive, FHI Solutions, Washington, District of Columbia, USA

**Contributors** PHN: conceived paper, analysis, drafted manuscript, consolidated comments from all co-authors, revised and finalised paper. SK: field work coordination, literature review, drafted some parts of the manuscript, revised and finalised paper. AP: field work coordination, data analyses, drafted some parts of the manuscript, reviewed manuscript. LMT: data analyses, visualisation for data presentation, reviewed the manuscript. SG, PKS, VDS and JE-A: data interpretation and its implications, reviewed and edited the manuscript. RA and PM reviewed the statistical analyses, supported data interpretation, reviewed and edited the manuscript. All authors read and approved the final submitted manuscript.

**Funding** Bill & Melinda Gates Foundation through POSHAN, led by International Food Policy Research Institute. Grant number: OPP50838.

**Competing interests** None declared.

**Patient consent for publication** Not required.

**Ethics approval** The research protocol received ethical clearance from the Institutional Review Board at the International Food Policy Research Institute (IRB #00007490) and the Suraksha Independent Ethics Committee in India (IRB #2017-10-9094). Additional permissions for data collection were provided by the State Government of Uttar Pradesh. Informed consent in the local language was obtained from mothers, frontline workers, and block managers prior to their participation in the study.

**Provenance and peer review** Not commissioned; externally peer reviewed.

**Data availability statement** All data relevant to the study are included in the article or uploaded as online supplemental information.

**ORCID iDs**
Phuong Hong Nguyen http://orcid.org/0000-0003-3418-1674

Vishal Dev Shastri http://orcid.org/0000-0002-5586-491X

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
