## [Reviewer comments · BMJ Open]

ARTICLE DETAILS

TITLE (PROVISIONAL)	The impact of COVID-19 on household food insecurity and interlinkages with child feeding practices and coping strategies in Uttar Pradesh, India: A longitudinal community-based study
AUTHORS	Nguyen, Phuong Hong; Kachwaha, Shivani; Pant, Anjali; Tran, Lan; Ghosh, Sebanti; Sharma, Praveen; Shastri, Vishal; Escobar-Alegria, Jessica; Avula, Rasmi; Menon, Purnima

VERSION 1 – REVIEW

REVIEWER	Pérez-Escamilla, Rafael Yale University, Global Health Leadership Institute
REVIEW RETURNED	13-Feb-2021

GENERAL COMMENTS	This longitudinal quantitative survey provides information on HFI trends among with 569 mothers with children in India. Dt are very relevant in the context of the COVID-19 pandemic because it was collected between December 2019 (in-person) and August 2020 (by phone). During this time HFI increased sharply from 21% to 80% before and during COVID-19 and difficult coping strategies needed to be implemented by families to deal with it. Analytical strategy to model changes in food insecurity status over time is appropriate. Manuscript is well written and has findings with key policy implications but needs to be improved taking the following recommendations into account. 1. Authors need to update their literature review to acknowledge and compare their findings to similar studies previously published. See these 3 articles as an example: (i) Food insecurity measurement and prevalence estimates during the COVID-19 pandemic in a repeated cross-sectional survey in Mexico. Public Health Nutr. 2021 Feb;24(3):412-421. doi: 10.1017/S1368980020004000; (ii) Social predictors of food insecurity during the stay-at-home order due to the COVID-19 pandemic in Peru. Results from a cross-sectional web-based survey. medRxiv [Preprint]. 2021 Feb 8:2021.02.06.21251221. doi: 10.1101/2021.02.06.21251221. (iii) Acute food insecurity and short-term coping strategies of urban and rural households of Bangladesh during the lockdown period of COVID-19 pandemic of 2020: report of a cross-sectional survey. BMJ Open. 2020 Dec 12;10(12):e043365. doi: 10.1136/bmjopen-2020-043365.2. Methods: Need to provide psychometric data to understand the validity (and comparability) of HFIAS in both modalities (in-person and phone). For phone version see Mexico study mentioned above.3. Lines 123-125. Please provide more details on specific procedures followed with 24-hour recall with mother. Was it a single pass or a multiple pass approach? Also were food models were used for portion sizes?, etc... If not, then perhaps you
---

	shouldn't refer to it as a 24 hour recall as that nomenclature has a very specific meaning in the dietary intake assessment methods. 4. Also why is dietary diversity relevant in this article since all infants in analytical sample were < 6 months (lines 129-31). Yet you do present dietary diversity data (see table 2). This is very confusing. 5. How was breast milk intake taken into account in dietary diversity index? 6. Results: Table 1 shows strong socio-economic differences between analytical sample and all mothers with data at baseline. Was this taken into account in analytical strategy? If so how? 7. Lines 319-20: See this article that presents a methodology to actually cost a cash transfer to protect families during COVID in Mexico: Costing of actions to safeguard vulnerable Mexican households with young children from the consequences of COVID-19 social distancing measures. Int J Equity Health. 2020 May 19;19(1):70. doi: 10.1186/s12939-020-01187-3..
--	---

REVIEWER	Okekunle, Akinkunmi Paul University of Ibadan
REVIEW RETURNED	13-Feb-2021

GENERAL COMMENTS	Nguyen et al have conducted a study on "The impact of COVID-19 on household food insecurity and interlinkages with child feeding practices and coping strategies in Uttar Pradesh, India" The study is promising and likely to provide hard evidence for the significance of the current COVID-19 pandemic on household food insecurity (HFI) and child dietary diversity. I would suggest the consideration of the manuscript for publication subject to the willingness of the authors in considering some issues I have thematically itemized in the attachment. - The reviewer provided a marked copy with additional comments. Please contact the publisher for full details.
--

VERSION 1 – AUTHOR RESPONSE

Reviewer: 1
Dr. Rafael Pérez-Escamilla, Yale University

Comments to the Author:

This longitudinal quantitative survey provides information on HFI trends among with 569 mothers with children in India. It are very relevant in the context of the COVID-19 pandemic because it was collected between December 2019 (in-person) and August 2020 (by phone). During this time HFI increased sharply from 21% to 80% before and during COVID-19 and difficult coping strategies needed to be implemented by families to deal with it. Analytical strategy to model changes in food insecurity status over time is appropriate. Manuscript is well written and has findings with key policy implications but needs to be improved taking the following recommendations into account.

Response: Thank you for the reviewer's comments on the strength of the paper.

1. Authors need to update their literature review to acknowledge and compare their findings to similar studies previously published. See these 3 articles as an example: (i) Food insecurity measurement and prevalence estimates during the COVID-19 pandemic in a repeated cross-sectional survey in Mexico. Public Health Nutr. 2021 Feb;24(3):412-421. doi: 10.1017/S1368980020004000; (ii) Social predictors of food insecurity during the stay-at-home order due to the COVID-19 pandemic in Peru. Results from a cross-sectional web-based survey. medRxiv [Preprint]. 2021 Feb

8:2021.02.06.21251221. doi: 10.1101/2021.02.06.21251221. (iii) Acute food insecurity and short-term coping strategies of urban and rural households of Bangladesh during the lockdown period of COVID-19 pandemic of 2020: report of a cross-sectional survey. *BMJ Open*. 2020 Dec 12;10(12):e043365. doi: 10.1136/bmjopen-2020-043365.

Response: We have now updated the literature in the introduction (lines 84-87, ref 13-18) and added the comparisons with previous studies in the discussion in (lines 274-281 and 282-285).

2. Methods: Need to provide psychometric data to understand the validity (and comparability) of HFIAS in both modalities (in-person and phone). For phone version see Mexico study mentioned above.

Response: We have included information on validity of measuring household food insecurity in-person vs. phone in lines 123-126.

3. Lines 123-125. Please provide more details on specific procedures followed with 24-hour recall with mother. Was it a single pass or a multiple pass approach? Also were food models were used for portion sizes?, etc... If not, then perhaps you shouldn't refer to it as a 24 hour recall as that nomenclature has a very specific meaning in the dietary intake assessment methods.

Response: Child feeding practices were assessed using the standard WHO indicators (WHO, 2008), on the basis of the maternal recall of all foods and liquids given to children in the 24 h prior to the survey. We did not conduct standard 24-hour recall. We have revised the text to avoid confusion (lines 133-135).

WHO. (2008). *Indicators for assessing infant and young child feeding practices. Part 2: Measurements*. http://whqlibdoc.who.int/publications/2010/9789241599290_eng.pdf. Retrieved from World Health Organization: Geneva

4. Also why is dietary diversity relevant in this article since all infants in analytical sample were < 6 months (lines 129-31). Yet you do present dietary diversity data (see table 2). This is very confusing.

Response: At the time of in-person survey in December 2019, all mothers had an infant between the ages of 0–5.9 months of age. In the follow-up phone survey in August 2020- children were 8-14 months. We have clarified this in lines 192-193.

5. How was breast milk intake taken into account in dietary diversity index?

Response: Breastfeeding was common, account for 91% of the sample. Based on WHO recommendation, breastmilk was not considered when constructing dietary diversity index, but we controlled for breastfeeding status in the model in Table 2.

6. Results: Table 1 shows strong socio-economic differences between analytical sample and all mothers with data at baseline. Was this taken into account in analytical strategy? If so how?

Response: We have now controlled for socio-economic status in all models (line 168). Most our findings remained similar.

7. Lines 319-20: See this article that presents a methodology to actually cost a cash transfer to protect families during COVID in Mexico: Costing of actions to safeguard vulnerable Mexican households with young children from the consequences of COVID-19 social distancing measures. *Int J Equity Health*. 2020 May 19;19(1):70. doi: 10.1186/s12939-020-01187-3..

Response: We have included information suggested in lines 337-339.

Reviewer: 2

Dr. Akinkunmi Paul Okekunle, University of Ibadan

General Comment

Nguyen et al have conducted a study on “The impact of COVID-19 on household food insecurity and interlinkages with child feeding practices and coping strategies in Uttar Pradesh, India” The study is promising and likely to provide hard evidence for the significance of the current

COVID-19 pandemic on household food insecurity (HFI) and child dietary diversity. I would suggest the consideration of the manuscript for publication subject to the willingness of the authors in considering some issues I have thematically itemized below.

Response: Thank you for the reviewer's comments to improve the paper.

Generally, the writing is good but can be improved. Employing the services of a professional English editor would not be out of place to address issues relating to grammatical errors, reporting in present tenses as against past tenses, the flow of expression, among others.

Response: We have done a thorough check to address any language issues.

The main challenge to the accuracy and scientific validity of this report is the use of a single measure of nutrition assessment – 24-hour dietary recall (24h-DR) in determining child feeding practices. This must be itemized in the discussion section of the manuscript as a limitation.

The ideological presupposition of using a 24hr-DR is quite inaccurate making it tedious to grasp the authors' conceptualization of this study. Using 24h-DR to define child feeding practices is inappropriate. This expression appears to have misrepresented the title and findings of the study. It is more appropriate to assert that the 24h-DR was used to determine child dietary diversity as against feeding practices. I am suggesting the authors consider revising the title and the entire manuscript to accurately reflect the true methods and results.

Response: Child feeding practices were assessed using the standard WHO indicators (WHO, 2008), on the basis of the maternal recall of all foods and liquids given to children in the 24 h prior to the survey. We did not conduct standard 24-hour recall. We have revised the text to avoid confusion (lines 133-135).

WHO. (2008). *Indicators for assessing infant and young child feeding practices. Part 2: Measurements.* http://whqlibdoc.who.int/publications/2010/9789241599290_eng.pdf. Retrieved from World Health Organization: Geneva

The result section was not well presented. For example, the use of "AOR" with no clear description of the meaning and absence of confidence interval is quite strange and confusing.

Response: We have revised the result section as suggested (lines 219-225, 243-246).

It is important to note that HFI has been a major public health problem in low- and middle income countries (LMIC) before the pandemic. It is a clear case of the weakness of the primordial spectrum of conceptual frame-work of malnutrition. However, it is unknown if the pandemic aggravated the HFI situation in LMIC. The authors' data would benefit the scientific literature in answering these questions.

Also, the authors are finding it tedious to discuss their findings in light of previously published reports. Agreeably, it is tedious to find pandemic-related studies in HFI to discuss their findings. However, situations of HFI in some LMIC in crises period such as economic recession, conflict, etc. could assist the authors in discussing their findings.

Response: We have now added the comparisons with previous studies in the discussion in lines 274-285.

Major Comments

Abstract

1. Any statistical power for consideration in the study? Also, any P-value for discerning statistical significance in the study?

Response: Statistical significance was defined as p value < 0.05. We have added this in the method (lines 170-171)

2. Line 33-34; The results have not been well presented.

"HFI increased sharply from 21% to 80% before and during COVID-19, with 62% households changing the status from food security to insecurity and 17% remaining food insecure."

"before and" and "insecurity and 17% remaining food insecure." should be removed.

Response: We have revised the sentence as suggested (lines 38-39).

3. What is the meaning of AOR?

Response: AOR means adjusted odd ratios – these result from multivariable regression models, adjusting for child age and sex, mother’s age, religion, and education, scheduled caste, number of children <5y in the household, and household social economic status. We have added adjusted factors in lines 36-27.

4. Line 39-40; The AOR are not well presented.

Response: We have revised the AOR presentation with 95% CI in lines 41-45.

5. Line 41; The first statement should be revised to reflect the findings of the study. It is tricky to assert “COVID-19 had a significant negative impact on HFI” when other factors have not been accounted for. It is quite true that the pandemic has a role to play in HFI but is plectonemically shrouded around multi-factorial issues primarily limited to lack of access (not only decorated in finances but also in lack of planning, instability in the value-chain of food production, etc.). The afore-mentioned is likely to vary across the spectrum of respondents.

Response: We have revised the sentence in line 47 to reflect the findings of the study.

Introduction

6. The Introduction is well written. However, there are a few grammatical errors that the authors should consider revising by seeking the assistance of a professional English editor. Also, the last statement in the introduction section in my opinion should not be an objective. The use of digital data collection tools predates the pandemic and it is not anything new to epidemiological surveys. In my opinion, that statement is not necessary.

Response: We have checked and corrected the language errors. We also removed the last statement in the introduction as suggested.

Materials and methods

7. Line 97-98; “Details of the parent study have described elsewhere” Please include the citation for the details of the parent study. Also, it is tedious to verify whether the parent study considered pandemic-related factors in the design. That could be a limitation to the design.

Response: We have included the citation for the details of the parent study (ref 24, line 105). The parent study was conducted before the outbreak of the pandemic, therefore, it could not consider pandemic-related factors in the design.

8. The use of acronyms was inconsistent. First, acronyms should be well defined at first mention. For example, what is the meaning of “HFI” – line 148, “FLW” – line 163, etc. Second authors should consider using acronyms consistently from the beginning of the manuscript to the end and not haphazardly.

Response: We have checked and used the acronyms consistently from the beginning of the manuscript to the end. HFI is defined at first use as household food insecurity in line 67-68.

9. Line 151 to 156 should be revised for clarity. The definitions of the categories appear clumsy and unclear. Again, any categorization or analysis to offer insight into food insecurity dynamics before the pandemic?

Response: We have revised the text describing the categories of food insecurity in lines 162-164. We used descriptive analysis to report HFI before and during the pandemic (line 159) and presented results in Figure 2.

10. Data analysis: any statistical power for consideration in the study? Also, any P-value for discerning statistical significance in the study?

Response: Statistical significance was defined as p value < 0.05. We have added this in lines 170-171.

11. 24-hour dietary recall: This method of nutritional assessment has a significant limitation. The time point of assessment is once and cannot be representative of the overall feeding pattern of the population. This limitation should be itemized in this manuscript.

Response: Child feeding practices were assessed using the standard WHO indicators (WHO, 2008), on the basis of the maternal recall of all foods and liquids given to children in the 24 h prior to the

survey. We have added the limitation of this nutritional assessment method in the discussion (lines 133-135).

Results

12. Line 174-181. I would expect a clear comparison of the characteristics of the analytic sample and non-analytic sample. I will advise the authors consider revising this section to itemize difference(s) in these population for the readers understanding.

Response: We have clarified the differences among analytic sample and non-analytic sample in lines 188-189.

13. Table 1: Statistically speaking the populations from the analytic sample and non-analytic sample appear statistically different. For example, years of school differed. The analytic sample had longer years of schooling. Also, household socio-economic status different. This is likely to confound the result of the study. Any statistical adjustment to resolve this? Otherwise, this is a major limitation.

Response: We have re-run the model, adjusted for education and household socio-economic status (lines 167-168). We also added this in the footnote of Tables 2 and 3.

14. Figures 2 & 4 have more than one figure embedded in them. It would be suitable the authors consider stressing which results are applicable for discussion in the result section. Generalizing the figure makes it unclear for readers to identify and link statistics cited in the results with the appropriate figure.

Response: We have revised Figure 2 to highlight applicable results for discussion.

15. Line 187-195. Please consider revising to appropriately cite the exact figure in the text.

Response: We have revised as suggested in lines 200-202.

16. Line 205-213: The authors keep "AOR" without clearly itemizing the meaning and perhaps the confidence interval (CI). It is a conventional practice to present odds and 95% CI. Where are the confidence intervals?

Response: We have added the confidence interval as suggested (lines 219-224 and 243-246).

17. Table 2: Any difference between the crude and the adjusted odds? It appears the crude odds are missing

Response: We have included the crude odds ratio in Tables 2-3 as suggested. The magnitude of association marginally attenuated in the adjusted model.

18. Line 222-226: No clear itemization of Tables and/or Figures where the results cited can be located.

Response: We have included the information in Figure 4 (new figure).

Discussion

19. The authors did well in itemizing the primary findings of the study but limited in discussing its findings in the light of similar settings. Authors should consider discussing their findings in the light of other LMIC.

Response: We have now added the comparisons with previous studies in the discussion in lines 274-285.

VERSION 2 – REVIEW

REVIEWER	Pérez-Escamilla, Rafael Yale University, Global Health Leadership Institute
REVIEW RETURNED	09-Mar-2021
GENERAL COMMENTS	Thanks for thoughtfully addressing all of my queries. Manuscript has been much improved and it is now clear what its innovative

	contributions are, and how it relates to other studies addressing household food insecurity in the context of COVID-19 in other parts of the world.
--	---

REVIEWER	Okekunle, Akinkunmi Paul University of Ibadan
REVIEW RETURNED	19-Mar-2021

GENERAL COMMENTS	Nguyen et al have revised their manuscript but there are concerns yet to be addressed.  1. The quoted line numbers in the response to the reviewer file cannot to reconciled with the manuscript file. Also, at some point in the manuscript, there were no line numbers. This made is significantly tedious to follow the revisions thematically and tie the efforts of authors with the outcome of the revision. 2. Line 46 should be revised. COVID-19 cannot be reported to pose a risk to HFI. Rather the pandemic and lockdown measures arising from the COVID-19 pandemic poses a threat to HFI. The same for the conclusion section of the manuscript. 3. Line 133-136: A supplementary table indicating food items from the 24hour DR that was used to derive the 7 groups may be necessary. 4. There are limitations worth mentioning to assist readers of this study. The authors should consider a paragraph at the end of the discussion to itemize these limitations. Just to mention a few;  (a) a single point 24-hour dietary recall for food/diet assessment may be unrepresentative of overall dietary exposure (b) residual confounding or unmeasured factors are potential factors to consider in examining our results. (c) whether the increase in the level of HFI (as a result of the measures promoted to manage the pandemic) affected child growth was not assessed in this study. This likely to lend credence to evidence for how HFI affects growth. This limitation is worth mentioning for future recommendations. (d) most coping measures for managing HFI in the study are related to managing expenditure. The HFI is likely to be more related to the threat to economic and earning power of household (as a result of measures imposed to manage the COVID-19 pandemic) than to the pandemic. (e) sequel to above, are there recommendations the author may suggest to guide threat against HFI in future pandemics. Going by the findings of this study. In a bid to prevent a pandemic, is it likely that another pandemic (food insecurity) is being created? Thank you
--

VERSION 2 – AUTHOR RESPONSE

Reviewer Reports:

Reviewer: 1

Dr. Rafael Pérez-Escamilla, Yale University Comments to the Author:

Thanks for thoughtfully addressing all of my queries. Manuscript has been much improved and it is now clear what its innovative contributions are, and how it relates to other studies addressing household food insecurity in the context of COVID-19 in other parts of the world.

Response: Thank you very much for your positive comments about our revised manuscript.

Reviewer: 2

Dr. Akinkunmi Paul Okekunle, University of Ibadan Comments to the Author:

Nguyen et al have revised their manuscript but there are concerns yet to be addressed.

1. The quoted line numbers in the response to the reviewer file cannot to reconciled with the manuscript file. Also, at some point in the manuscript, there were no line numbers. This made is significantly tedious to follow the revisions thematically and tie the efforts of authors with the outcome of the revision.

Response: We used the continuous line numbers that correspond to double spacing of the manuscript. When submitted, the website automatically generates other line numbers for the single space, and for each page separately- which cause the confusion. We plan to provide this feedback to editors- to they can improve the submission site in the future.

2. Line 46 should be revised. COVID-19 cannot be reported to pose a risk to HFI. Rather the pandemic and lockdown measures arising from the COVID-19 pandemic poses a threat to HFI. The same for the conclusion section of the manuscript.

Response: We have revised the conclusion of abstract (line 47) and the paper (line 367) as suggested.

3. Line 133-136: A supplementary table indicating food items from the 24hour DR that was used to derive the 7 groups may be necessary.

Response: We have now provided a supplementary table (Supplemental Table 1, line 134) indicating food items that was used to derive the 7 groups.

4. There are limitations worth mentioning to assist readers of this study. The authors should consider a paragraph at the end of the discussion to itemize these limitations. Just to mention a few;

(a) a single point 24-hour dietary recall for food/diet assessment may be unrepresentative of overall dietary exposure

(b) residual confounding or unmeasured factors are potential factors to consider in examining our results.

(c) whether the increase in the level of HFI (as a result of the measures promoted to manage the pandemic) affected child growth was not assessed in this study. This likely to lend credence to evidence for how HFI affects growth. This limitation is worth mentioning for future recommendations.

(d) most coping measures for managing HFI in the study are related to managing expenditure. The HFI is likely to be more related to the threat to economic and earning power of household (as a result of measures imposed to manage the COVID-19 pandemic) than to the pandemic.

(e) sequel to above, are there recommendations the author may suggest to guide threat against HFI in future pandemics. Going by the findings of this study. In a bid to prevent a pandemic, is it likely that another pandemic (food insecurity) is being created?

Response: We have added the limitations of the study in lines 359-364 and also provided recommendations in lines 375-378.

VERSION 3 – REVIEW

REVIEWER	Okekunle, Akinkunmi Paul University of Ibadan
REVIEW RETURNED	04-Apr-2021

GENERAL COMMENTS

I am of the opinion the current version of the manuscript reflects a balanced and well-informed synopsis of the author's study. I have no further comments